# Observation of an antiferromagnetic quantum critical point in high-purity LaNiO₃

Changjiang Liu[1], Vincent F.C. Humbert[2], Terence M. Bretz-Sullivan[1], Gensheng Wang[3], Deshun Hong[1], Friederike Wrobel [1], Jianjie Zhang[3], Jason D. Hoffman[4], John E. Pearson[1], J. Samuel Jiang[1], Clarence Chang[3], Alexey Suslov [5], Nadya Mason[2], M.R. Norman[1] & Anand Bhattacharya [1✉]

Amongst the rare-earth perovskite nickelates, LaNiO₃ (LNO) is an exception. While the former have insulating and antiferromagnetic ground states, LNO remains metallic and non-magnetic down to the lowest temperatures. It is believed that LNO is a strange metal, on the verge of an antiferromagnetic instability. Our work suggests that LNO is a quantum critical metal, close to an antiferromagnetic quantum critical point (QCP). The QCP behavior in LNO is manifested in epitaxial thin films with unprecedented high purities. We find that the temperature and magnetic field dependences of the resistivity of LNO at low temperatures are consistent with scatterings of charge carriers from weak disorder and quantum fluctuations of an antiferromagnetic nature. Furthermore, we find that the introduction of a small concentration of magnetic impurities qualitatively changes the magnetotransport properties of LNO, resembling that found in some heavy-fermion Kondo lattice systems in the vicinity of an antiferromagnetic QCP.

---

[1] Materials Science Division, Argonne National Laboratory, Lemont, IL 60439, USA. [2] Department of Physics, University of Illinois at Urbana-Champaign, Urbana, IL 61801, USA. [3] High Energy Physics Division, Argonne National Laboratory, Lemont, IL 60439, USA. [4] Department of Physics, Harvard University, Cambridge, MA 02138, USA. [5] National High Magnetic Field Laboratory, Tallahassee, FL 32310, USA. ✉email: anand@anl.gov

n the vicinity of a quantum phase transition (QPT) near $T = 0$ K, a material can be tuned in and out of an ordered state using a parameter other than temperature, such as magnetic field or pressure[1,2]. Near a QPT, quantum fluctuations of the order parameter can have profound influence on the properties of the material, out to high temperatures[3]. For example, in a metal, quantum fluctuations can introduce long-range interactions between mobile electrons causing a breakdown of the Landau Fermi liquid (LFL)[4], leading to a strange metal with anomalous transport and thermodynamic properties[5]. Quantum fluctuations can also mediate superconducting pairing of carriers and give rise to unusual responses to electric and magnetic fields. In rare instances, a material can be found to be intrinsically quantum critical, perched on the edge of a QPT without the need for tuning. Here we report on signatures of an antiferromagnetic (AFM) QCP in high-purity LaNiO$_3$ thin films. We find that the resistivity $\rho(T)$ shows a linear temperature dependence over almost a decade of $T$ below ~1.1 K in our cleanest samples. The linear-in-$T$ resistivity crosses over to a $T^2$ dependence in a magnetic field, consistent with the presence of AFM quantum critical fluctuations.

The rare-earth nickelates ($Re$-NiO$_3$) are a widely studied family of materials that display rich electronic and magnetic properties as a result of competition between itinerancy and electron–electron and electron–lattice interactions that tend to localize carriers and give rise to an AFM insulating ground state. They have a perovskite structure, with Ni cations at the center of corner-sharing $O$ octahedra. For smaller $Re$ cations[6,7], an insulating state is obtained at temperatures below $T_{MIT}$ (metal–insulator transition temperature), accompanied by a structural distortion. At yet lower temperatures $T_N \leq T_{MIT}$ ($T_N$ is the Néel temperature), an AFM state is obtained. As the $Re$ cation radius increases, $T_{MIT}$ decreases, merges with $T_N$, and eventually both are driven to zero. LaNiO$_3$ (LNO), with the largest $Re$ cation, is the only $Re$-NiO$_3$ nickelate that is metallic down to the lowest temperatures. LNO is a correlated metal[8] and it has long been suspected that the properties of LNO are influenced by its proximity to magnetic and structural instabilities[9,10], perhaps even by quantum fluctuations resulting from these instabilities[11,12]. However, clear experimental evidence for the effect of quantum fluctuations at low temperatures has not been reported in LNO until now.

Here, we report on systematic study on a series of LNO samples with varying degrees of disorder. The dependence of the resistivity on temperature shows agreement with theories that consider the interplay between scattering from disorder and quantum AFM fluctuations. Furthermore, we observe signatures of spin-flip scattering from localized spins in samples with greater levels of disorder at higher temperatures, while at low temperatures signatures of AFM correlations emerge, similar to behavior observed in some heavy-fermion systems. Our findings indicate that the low-temperature transport properties of LNO arise from the interplay between an AFM quantum critical point, impurity scattering and the interaction of itinerant carriers with localized spins.

## Results

### Sample preparation
Epitaxial LNO thin films are grown on (001) oriented (LaAlO$_3$)$_{0.3}$(Sr$_2$AlTaO$_6$)$_{0.7}$ (LSAT) substrates by ozone-assisted molecular beam expitaxy. The growth parameters for controlling the La/Ni ratio are determined from Rutherford backscattering spectrometry (RBS) measurement (see Supplementary Fig. 1). To ensure that the oxygen vacancies are minimized, we used a high ozone flux with a background pressure of $7 \times 10^{-6}$ torr. The growth process was monitored by reflection high-energy electron diffraction (RHEED) (see Supplementary Fig. 2). Details of the sample growth, the control of stoichiometry and X-ray characterizations are presented in "Methods" and Supplementary Fig. 3. Transport measurements were performed on six-terminal devices patterned in Hall bar geometry using photolithography. Seven samples are studied in this work, and they are labeled as LNO_#, with # being close to the residual resistivity ratio [RRR = $\rho$(300 K)/$\rho$(2 K)] of the sample. The disorder level in the sample may be characterized approximately by the residual resistivity or RRR of the sample (note that Matthiessen's rule might not apply at $T = 2$ K due to an interplay between disorder and other scattering mechanisms).

### Resistivity measurement in the high-purity sample
Figure 1a shows the temperature dependence of the resistivity measured on sample LNO_24. This sample shows a resistivity of 3.8 μΩ cm and a mobility of about 160 cm$^2$ V$^{-1}$ s$^{-1}$ at $T = 2$ K. The corresponding RRR is about 24, which is so far the highest RRR reported for LNO (see Supplementary Note 1 and Table 1). When the transport measurement was extended down to 25 mK, the resistivity continues to decrease without showing any flattening until about 100 mK, and in fact it shows an unexpected linear-in-temperature dependence in the temperature range 0.1 K < $T$ < 1.1 K (Fig. 1b). The same behavior in the resistivity is observed in multiple samples (see Supplementary Fig. 4). A description of the uncertainties of the data points is presented in "Methods". Phonons are nominally not relevant in this low-temperature regime given that the Debye temperature of LNO is above 400 K[13]. Ni–O bond length fluctuations could be a source of linear—$T$ resistivity at low temperatures[14], and there may be other mechanisms as well (charge fluctuations, Umklapp scatterings). However, these mechanisms are less natural for explaining our data as they would have a weak dependence on magnetic field, which will be discussed in the following. This behavior of the resistivity in LNO is in sharp contrast with the LFL theory, which predicts a quadratic temperature dependence of resistivity at low temperatures. In Fig. 1c, we plot the resistivity exponent $\alpha$ (solid line) in $\rho(T) = \rho_0 + AT^\alpha$, as a function of temperature. Here $\rho_0$ is the residual resistivity, and $\alpha$ is calculated from $d \ln(\rho - \rho_0)/d \ln(T)$. Before taking derivatives, the raw data were first smoothed using a B-Spline method. We find that $\alpha$ initially oscillates around a value of 1.5 from $T = 300$ K to about 30 K (Fig. 1c). As temperature decreases further, $\alpha$ first increases to a value of about 1.75 near $T = 7.5$ K and then begins to decrease and approaches a constant value of ~1 in the sub-Kelvin regime.

Previous studies found that the resistivity of LNO shows a $T^{1.5}$ power law at temperatures above about 30 K[8,13,15], and anomalous exponents have been observed in other nickelates[16,17]. Similar behavior is observed here as we saw in Fig. 1c (see also Supplementary Fig. 5). A $T^{1.5}$ power law scaling is often attributed to scattering by AFM spin fluctuations, as proposed by the Hertz–Millis–Moriya model for systems close to an AFM QCP[18–20]. Electronic states on the Fermi surface connected by wave vectors of the incipient AFM ordering (hot regions in $k$-space) are strongly scattered. However, these hot regions only occupy a finite phase space on the Fermi surface, and in the clean limit they are shorted out by the cold regions where such scattering does not operate[21]. It was realized subsequently that the scattering due to quantum AFM spin fluctuations can lead to a peculiar evolution[22,23] of $\alpha$ with $T$ that depends on the level of disorder. For systems close to AFM QCP in the dirty limit, a $T^{1.5}$ power law should be observed over a large range of temperature. For cleaner systems ($k_F l > 10$, where $k_F$ is the Fermi wavevector and $l$ is the mean free path), as temperature decreases, $\alpha$ first increases and approaches values close to 2 (symbols in Fig. 1c), as

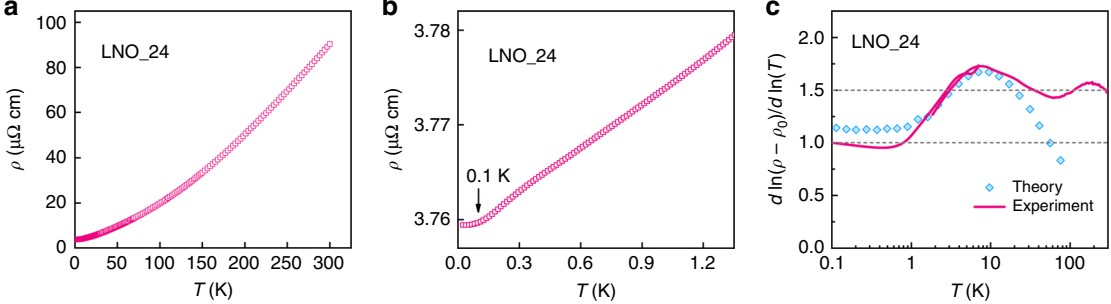

**Fig. 1 Temperature dependence of resistivity and evolution of the resistivity exponent. a** Resistivity of LNO measured as a function of temperature from 300 K to 2 K. **b** Linear-in-temperature resistivity observed at temperatures below about 1.1 K. **c** Temperature dependence of the resistivity exponent $\alpha$ (solid line) from 300 K to 100 mK. The two horizontal dashed lines indicate exponent values of 1 and 1.5, respectively. Symbols show theoretical predictions of the resistivity exponent $\alpha$ for a clean sample ($k_F l \sim 1000$) at AFM QCP.

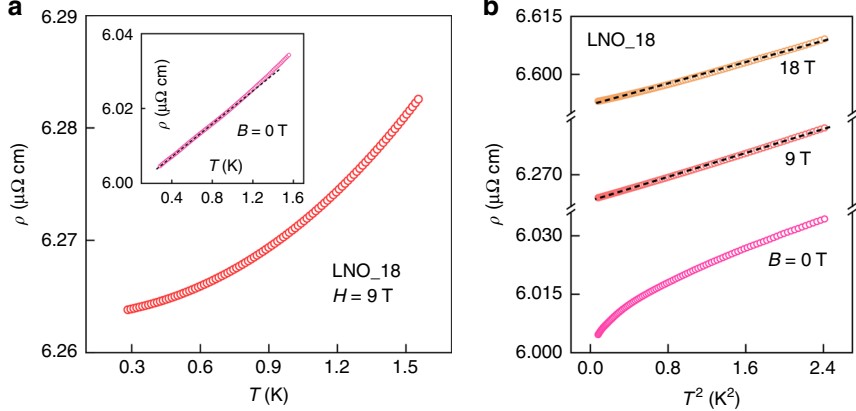

**Fig. 2 Temperature dependence of resistivity under a magnetic field. a** Resistivity measurement under a magnetic field of 9 tesla for LNO_18. Inset shows the linear-in-temperature behavior under zero field. **b** Temperature dependence of resistivity plotted versus the square of temperature for different magnetic fields. The measurements at $B = 9$ and 18 tesla show a linear dependence on $T^2$. Note the positive magnetoresistance. The standard deviation of the mean at each point is smaller than the symbol size. The black dashed lines in **b** and the inset of **a** are straight guidelines.

expected for LFL quasiparticles. At lower temperatures, $\alpha$ decreases continuously to a value ~1. A signature characteristic of this model is a bump in $\alpha$ as a function of $T^{22}$, which is clearly observed here at temperature $T_{Bump} \sim 7.5$ K (Fig. 1c). At even lower temperatures T $< 10^{-3}T_{Bump}$, which are not experimentally accessible for us, $\alpha$ increases again to the dirty limit value of 1.5 according to this theory. Using the resistivity and Hall measurements (see Supplementary Fig. 6), we estimate $k_F l \sim 500$ for our cleanest samples at $T = 2$ K. Symbols in Fig. 1c are calculations reproduced from ref. [22] for a clean sample with $k_F l \sim 1000$ (see also Supplementary Fig. 7). Further details of the analysis are presented in Supplementary Note 2.

**Restoration of LFL under a magnetic field**. If AFM fluctuations near a QCP are responsible for the linear $\rho(T)$ that we observe at low temperatures, an applied magnetic field may be used to suppress these fluctuations and restore the LFL. Such behavior has been observed in heavy-fermion compounds close to a QCP —for example in YbRh$_2$(Si$_{1-x}$Ge$_x$)$_2$ and CeCoIn$_5$[24,25]. Figure 2a shows our measurement results on LNO_18 under a magnetic field of 9 T, presenting a quadratic dependence of resistivity on temperature, i.e., $\rho(T) = \rho_0 + AT^2$. Inset of Fig. 2a shows the linear $\rho(T)$ in zero magnetic field in the sub-Kelvin regime, as has been observed in other clean samples. Thus, a restoration of LFL behavior in LNO is observed under a magnetic field. This is seen more clearly in Fig. 2b, where we plot $\rho(T)$ vs $T^2$ at different

fields. The data can be fit well to a $T^2$ dependence for $B = 9$ and 18 T, with corresponding $A$ coefficients in $\rho(T)$ being $8 \times 10^{-3}$ and $7 \times 10^{-3}$ $\mu\Omega$ cm K$^{-2}$, respectively. These values of $A$ are about 3–4 times larger than those, ~$2 \times 10^{-3}$ $\mu\Omega$ cm K$^{-2}$, reported for LNO previously[13,15,26], which is suggestive of enhanced scattering near the QCP. We note that this crossover to LFL in the resistivity under a magnetic field is seen only in samples with RRR $\geq 18$. Therefore, the presence of disorder in LNO could detune the system away from the QCP.

**Interplay of AFM spin fluctuations and disorder**. To examine the role of disorder on the quantum critical behavior of LNO, we grew a series of samples under different ozone pressure, substrate temperatures, and with slight variance in La/Ni ratio (<2%). The growth conditions of each sample are presented in Supplementary Table 2. The main form of impurities introduced during the growth are oxygen vacancies and Ni$^{2+}$, which also act as local magnetic moments. The sample with the highest RRR was grown under the highest effective ozone pressure, a substrate temperature of 615 °C, and a La/Ni ratio very close to 1. Figure 3 shows the resistivity measurement on these samples. As sample become cleaner (increasing RRR from left to right), there is a corresponding change in the temperature range (shaded region, from $T_{low}$ to $T_{high}$) over which a linear $\rho(T)$ appears. In cleaner sample, $T_{low}$ decreases, while the temperature ratio $T_{high}/T_{low}$ over which we observe linear $\rho(T)$ increases. This behavior is consistent with

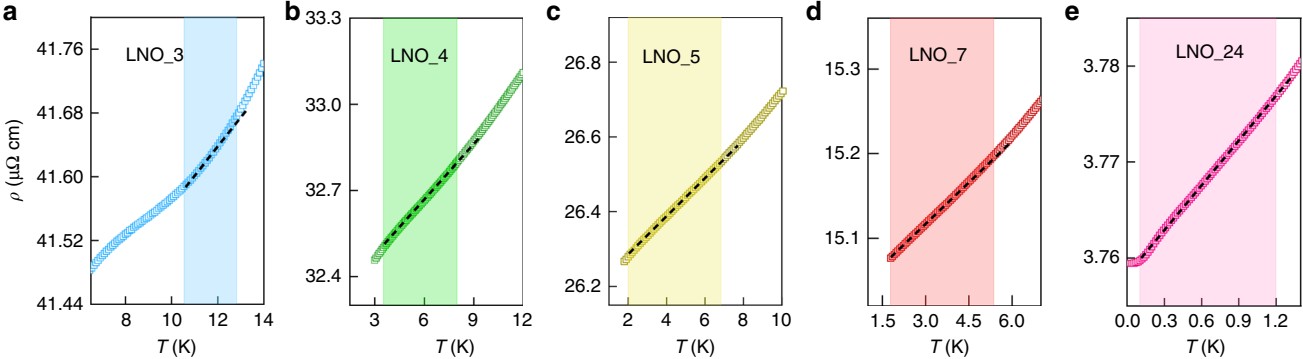

**Fig. 3 Resistivity of samples with different disorder levels. a–e** Low-temperature measurement of resistivity on samples with different impurity levels. The shaded region in the plot indicates the temperature range where the resistivity shows a linear temperature dependence (quasilinear for low-RRR samples). The black dashed lines are straight guidelines. A downward curvature in the resistivity is observed in these samples near the low-temperature end $T \leq T_{low}$.

the Rosch's model[23] where $T_{low} \sim 1/k_F l$ and $T_{high}/T_{low} \sim \sqrt{k_F l}$ (see Supplementary Note 2).

We found that at lower temperatures ($T \leq T_{low}$), the resistivity shows anomalous sublinear temperature dependence. This is identified as a downward curvature in the resistivity curve, which is clearly seen in Fig. 3a at $T < 10$ K. Similar behaviors in the resistivity have been observed in heavy-fermion Kondo lattice systems. In CeCo$_x$Rh$_{1-x}$In$_5$ compounds, the appearance of a sublinear $\rho(T)$ was associated with the formation of short-range AFM order together with Kondo coherence, which are intimately related to an AFM QCP. Short-range AFM order may also emerge in LNO at low temperatures. This is corroborated by measurements in a magnetic field, which suppresses weak AFM order. With an applied magnetic field of 9 T, the sublinear part of $\rho(T)$ at low temperatures is mostly quenched (see Supplementary Fig. 8a). As the sample's RRR becomes higher, both the magnitude and the onset temperature of the sublinear part of $\rho(T)$ becomes smaller. In our cleanest sample (shown in Fig. 3e or Fig. 1b), a small sublinear $\rho(T)$ can be identified only at very low temperatures (<300 mK), suggesting that pure LNO is in close proximity to an AFM QCP. We note that a small sublinear $\rho(T)$ has also been observed at low temperatures in high-quality bulk single-crystal LNO (Supplementary Fig. 8b).

**Scattering from local magnetic moments.** According to current understanding, nickelates are negative charge-transfer materials[7,27,28] where the nominally $3d^7$-states of the Ni$^{3+}$ sites take one electron from oxygen leading to a $3d^8\underline{L}$ electron configuration, where $\underline{L}$ stands for a ligand hole on the O $2p$ orbitals. This may be seen (approximately, due to Ni–O covalence) as a localized $3d^8$ magnetic moment ($S = 1$) at a Ni impurity site[29] that is partially screened by the ligand hole which takes on the $e_g$ symmetry of the $3d^8$ orbitals. The Fermi level lies in the strongly hybridized O $2p$ band, with the lowest lying excitations being from $2p$ (filled) to $2p$ (empty) states in the continuum. Such low lying excitations have been inferred[30] from resonant X-ray inelastic scattering measurements in NdNiO$_3$, and the dominance of holes in magnetotransport and thermogalvanic properties has also been experimentally demonstrated[31] recently for LNO. The screening and hybridization between the conduction charge carriers and localized magnetic moments in LNO bear resemblance to an underscreened Kondo lattice[32]. Thus, when an electron is doped into LNO, it would fill a ligand hole $\underline{L}$ and create an extra unscreened $3d^8$ moment localized on the Ni site, which can act like a magnetic scattering center.

In our samples, we observe a systematic evolution of the magnetoresistance with impurity levels. Figure 4a shows our longitudinal magnetoresistance (LMR) measurement results. The magnetic field is in the film plane and parallel to the current direction, which minimizes orbital MR effects from the Lorentz force. The LMR changes gradually from negative to positive as the RRR increases from 3.3 (LNO_3) to 24 (LNO_24). In samples, such as LNO_3, showing a large negative LMR, the magnitude of the LMR does not depend on the direction of the magnetic field (see Supplementary Fig. 9). The resistivity also shows metallic behavior down to the lowest temperature. These phenomena could not be explained by a weak localization mechanism[33]. Furthermore, for $T \geq 10$ K, the normalized MR defined as $\Delta\rho/\rho = [(\rho(B,T) - \rho(0,T)]/\rho(0,T)$, can be made to collapse by plotting $\Delta\rho/\rho$ vs. $B/(T + T')$, allowing for a variance ($\pm 2$ K) in the free parameter $T'$, as shown in Fig. 4b (note that $T' \ll T$ in this temperature range). In particular, $\Delta\rho/\rho \sim -[B/(T + T')]^2$, in agreement with predictions for negative MR induced by spin-flip scattering from localized impurities[34]. Here, the overall $B/(T + T')$ scaling can be understood as the temperature dependence of the magnetic susceptibility $\chi$ of single ions; $T'$ is a measure of correlation among them, analogous to the Curie–Weiss temperature. At lower temperatures ($T < 7$ K), shown in Fig. 4c, while $\Delta\rho/\rho \sim -B^2$ still holds, the negative LMR begins to saturate, and $T'$ increases as $T$ decreases approximately as $T' \sim (6.2\ K - T)$. Essentially, the denominator in $B/(T + T')$ remains constant at low temperatures. This behavior is similar to the Curie–Weiss law for an antiferromagnet, where the magnetic susceptibility remains finite when the AFM correlation increases at low temperatures (see Supplementary Figs. 10 and 11). Crucially, the temperature at which the negative LMR starts to saturate matches the temperature where the $\rho(T)$ becomes sublinear. These magnetotransport measurements, therefore, further support that pure LNO is in the vicinity of an AFM QCP; while the introduction of magnetic impurities results in short-range AFM ordering (see Supplementary Note 3). It has been shown that only a small amount of oxygen vacancies in LNO, which promotes the formation of Ni$^{2+}$ sites, can result in long-range magnetic ordering[35].

## Discussion

The transport property of LNO shows similarities to those reported for Kondo lattice or heavy-fermion materials[36]. For instance, CeCoIn$_5$ shows a linear $\rho(T)$ at low temperatures, which becomes quadratic in a magnetic field. The magnetic field dependence of resistivity also shows a $\Delta\rho/\rho \sim -[B/(T + T')]^2$

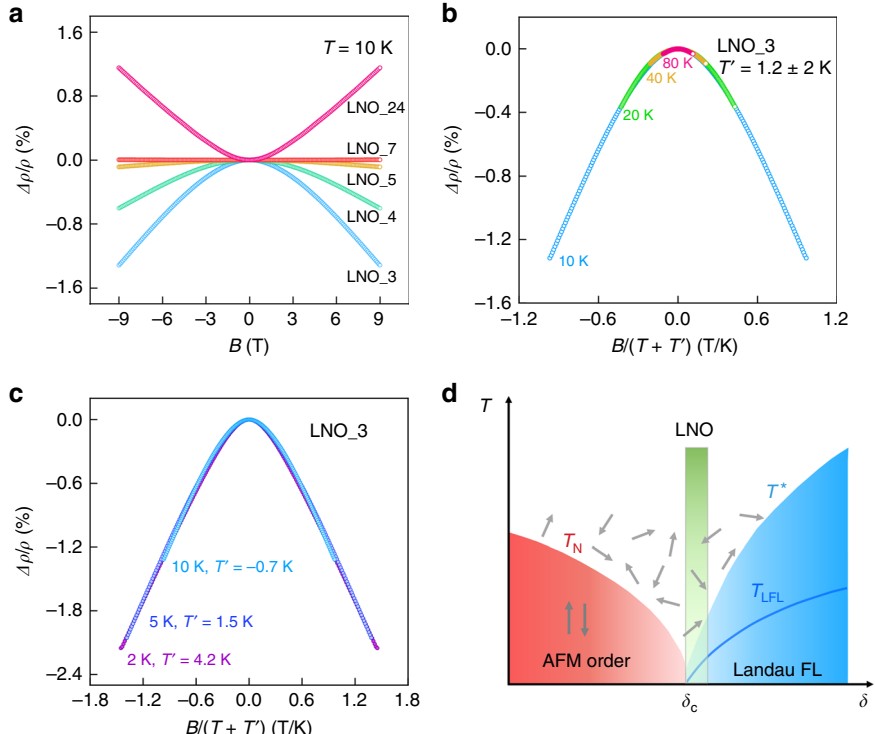

**Fig. 4 Magnetotransport measurements and phase diagram. a** Longitudinal magnetoresistance measurements for a series of samples at 10 K. **b**, **c** The negative LMR measured on LNO_3 at temperatures above and below 10 K, respectively. The x-axis is scaled by $B/(T + T')$ as discussed in the text. **d** Phase diagram of a system with Kondo lattice character. Green rectangle represents the position where LNO is likely to sit. $T^*$ marks the temperature at which Kondo coherence starts to develop. $T_{LFL}$ represents the temperature below which LFL behavior appears, such as a quadratic temperature dependence of the resistivity. Small arrows with random orientations represent local magnetic moments.

dependence. Thus, the low-temperature physics of LNO might involve a subtle interplay of AFM quantum fluctuations and Kondo physics[37–40]. Figure 4d shows a proposed phase diagram containing LNO. As the tuning parameter $\delta$ increases, the system undergoes a transition from an AFM phase (under $T_N$) to an LFL state. Here, $\delta$ can be the Re cation radius or strain. The green rectangle represents where LNO is likely situated in this phase diagram, near the critical point and on its right side. It is understood from this phase diagram that LNO can be driven from strange metal towards an LFL by an external magnetic field upon suppressing AFM fluctuations near the QCP. At elevated temperatures, local magnetic moments give rise to single-impurity spin-flip scattering, particularly for samples with a higher degree of disorder. Recently, superconductivity has been observed in an infinite-layer nickelate[41], which might have a connection to the QCP that we observe here in LNO. This possibility warrants further exploration.

In summary, we have observed non-LFL behavior at low temperatures in high-purity epitaxial thin films of LNO. In particular, a linear-in-temperature dependence of resistivity is observed that extends over almost a decade of temperature at $T \leq 1.1$ K in our cleanest samples. The evolution of the resistivity exponent as a function of temperature follows model predictions for a system with three dimensional quantum critical AFM fluctuations and weak impurity scattering. These fluctuations are suppressed in a magnetic field, and the LFL is restored. These results, and the systematics of single-impurity scattering in samples with elevated disorder level suggest that high-purity LNO is on the verge of an AFM quantum phase transition.

## Methods

**Sample growth details**. When the effusion cells containing La and Ni sources are heated to deposition temperatures ( ~1430 °C for La and ~1290 °C for Ni), atoms

are evaporated toward the substrate at a stable rate. The deposition rate is measured by a quartz crystal microbalance (QCM) before and after growth. To calibrate the QCM measurement, we deposit La and Ni on a MgO substrate using the same growth condition as for the actual samples. We then use RBS to determine the relative ratio of La to Ni. Using a MgO substrate makes the background near La and Ni peaks clean in the RBS measurement. Shown in Supplementary Fig. 1 is the analysis of the RBS data, which give a ratio of La to Ni of about 1.009. This value was then used to adjust the shutter times of La and Ni sources during growth to target a nominal La/Ni ratio to as close to one as possible. In our lowest resistivity samples, the drift in La and Ni rates during growth were under 0.3% h$^{-1}$.

Supplementary Table 2 presents detailed growth parameters of the samples discussed in the main text. The growth temperature was measured from a thermocouple. Elemental sources of La and Ni were evaporated sequentially from effusion cells using a block-by-block technique. The layer sequence was LaO–NiO$_2$–LaO.... The growth process was monitored by RHEED. Shown in Supplementary Fig. 2a is the RHEED intensity measured as a function of time as growth proceeds. When the LaO layer is deposited the surface becomes rough, which results in a drop in RHEED intensity as shown in Supplementary Fig. 2b. The deposition of a NiO$_2$ layer makes the sample surface smooth enhancing the RHEED intensity, which is shown in Supplementary Fig. 2c. We note that RHEED intensity oscillations can have complex origins[42]. Therefore, one period of a RHEED oscillation corresponds to one unit-cell of LaNiO$_3$. In Supplementary Fig. 2c, half ordering peaks can be clearly identified, which indicates good surface crystallinity of the LaNiO$_3$ film. We observed that although the high-RRR samples have good surface crystallinity, the surface roughness is higher than those low-RRR samples grown at lower temperatures. Therefore, the sample's resistivity does not positively correlate with the sample's roughness. Supplementary Fig. 2d shows the characterization of the surface roughness on sample LNO_18 using X-ray reflectivity (XRR) measurement. The roughness of this high-RRR sample is about 0.86 nm, which is higher than that of LNO_3 of about 0.38 nm determined from the same measurement. After growth, the film's thickness and c-axis parameter were characterized by low-angle XRR and X-ray diffraction (XRD) measurements, respectively (see Supplementary Fig. 3). The film thickness obtained from fitting the XRR data was ~30.8 nm, which is in good agreement (<1% error) for a 80-unit cell LNO with the c-axis lattice constant of ~3.82 Å obtained from the XRD.

**Device fabrication and measurement details**. Standard photolithography was used to fabricate Hall bar devices used in the transport measurement. To prevent the formation of oxygen vacancies, liquid-nitrogen cooling was used during Ar-ion

milling. Electrical contacts were made by depositing 50-nm thick platinum using sputtering. The channel of the Hall bar has a dimension of $50 \times 800$ $\mu m^2$, and the distance between the two voltage contacts is 400 $\mu m$. Low-noise transport measurements were performed using a Lakeshore model 372 AC resistance bridge, Stanford lock-in amplifiers and Keithley 6221/2182A current source/nanovolt-meters in delta mode. Typical excitation currents of ~2 $\mu A$ were used in our measurements. Cooling of electrons to temperatures below 100 mK was achieved by multistage filtering of the measurement lines in a dilution fridge.

**Uncertainties of the measurement**. In all figures including those in the Supplementary Information, the data points for the resistivity measurement are interpolations or mean values of the raw data which have higher density than those shown in each plot. The error bar, representing the standard deviation of the mean at each data point, is smaller than the symbol size. In these measurements, the uncertainty in resistivity is ~$8 \times 10^{-4}$ $\mu\Omega$ cm. There is a systematic uncertainty (<10%) in resistivity that comes from variations in the dimensions of individual device, which does not affect the RRR value of the sample.

## Data availability

All data presented in the main text and all resistivity data in the Supplementary Information are available for download at the following url: https://doi.org/10.5061/dryad.zpc866t5v.

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

## Acknowledgements

All work at Argonne including film growth, characterization, and magnetotransport measurements were supported by the US Department of Energy, Office of Science, Basic Energy Sciences, Materials Sciences and Engineering Division including support for F.W. via the Center for Predictive Simulation of Functional Materials. The use of facilities at the Center for Nanoscale Materials, an Office of Science user facility, was supported by the US Department of Energy, Basic Energy Sciences under Contract No. DE-AC02-06CH11357. Dilution fridge measurements carried out by G.W., J.Z., and C.C. at the Argonne High-Energy Physics Division was supported by the US Department of Energy, Office of Science, High-Energy Physics. Dilution fridge measurements at UIUC were carried out by V.F.C.H. and N.M., supported by the National Science Foundation under NSF DMR- 1710437. Measurements at low temperatures and high magnetic fields were carried out at the NHMFL, which is supported by the NSF Cooperative agreement no. DMR-1644779 and the State of Florida. We thank Peter Littlewood, John Mitchell, and Andrew Millis for discussions.

## Author contributions

C.L. grew and characterized the LNO samples with assistance from D.H., J.D.H., and F. W. and J.E.P. Transport measurements were carried out by C.L., V.F.C.H., T.B.S., G.W., J.Z., J.S.J., C.C., A.S., and N.M. down to dilution fridge temperatures. M.R.N. provided theoretical guidance. C.L. and A.B. wrote the paper with contributions from all authors. A.B. supervised the project.

## Competing interests

The authors declare no competing interests.
