## [Peer Review File · Nature Communications]

Reviewers' comments:

Reviewer #1 (Remarks to the Author):

The major claim of the submitted manuscript is that LaNiO₃ is proximal to an antiferromagnetic quantum critical point. The evidence to support the conclusion consists of: (1) a linear temperature dependence of the resistivity observed over a certain range of temperatures in various samples; (2) the crossover to quadratic temperature dependence with magnetic field; (3) the scaling of the magnetoresistance with some function of B/T (magnetic field/temperature); (4) the agreement of the temperature evolution of the temperature exponent of the resistivity with a theoretical model for scattering due to quantum spin fluctuations.

Given recent debate about the nature of the intrinsic ground state in LaNiO₃ and in the context of recently reported superconductivity in nickelate structures, the paper addresses an important, timely, and relevant topic. The presented data are certainly suggestive of quantum critical behavior and the phenomenology resembles other systems that are known to be influenced by antiferromagnetic quantum fluctuations such as heavy fermion compounds. However, I find several technical issues with the discussion and have a few pertinent questions. Therefore, I might recommend the manuscript for publication in Nature Communications, but I think the authors should address these issues before proceeding. They are enumerated below:

(1) A key part of the discussion is related to the influence of disorder on the quantum critical signatures of LaNiO₃. The relative amount of disorder is presumably deduced from the RRR of different samples, and it is speculated that the disorder is introduced from off-stoichiometry, leaving isolated Ni impurities. But, from the sample table shown in the supplement, there seems to be little correlation between the La/Ni ratio of the film and the RRR. Rather, the RRR seems to be more correlated with differences in growth conditions such as growth temperature and ozone nozzle distance, which may point to differences in oxygen incorporation in the films. One may then suspect that oxygen vacancies contribute the main impurities. This will still likely lead to localized Ni impurities. In any case, the notion that the disorder was systematically varied by the growth conditions appears doubtful. The authors should comment on the form of disorder, which may not be particularly known, and what kind of impact this has on the results.

(2) While the range of temperature that shows purported linear behavior can be explained by increased disorder, it is not clear why the absolute temperature at which linear behavior arises should increase. What is the origin of this?

(3) Showing a line on top of a small region of temperature does not necessarily make a strong case for linear T behavior, and the claim that linear T behavior is seen over a decade is not obvious from the plot of Fig. 1(b), as the resistivity deviates from linear rather clearly starting around 0.7K or so. Can the authors show more definitive evidence for the linear T behavior? For example, the plot of the temperature exponent in Fig. 1(c) provides more valuable information. I suspect that a similar plot for the other samples would help clarify which truly show linear T behavior (and over which range) as well as for which samples the linear T region is just a crossover between the super- and sub-linear regimes.

(4) The discussion first begins with a clean sample which shows T linear behavior at low temperatures, then another one for which a high magnetic field restores T² behavior, then a group of samples of different disorder, then a disordered sample showing negative MR. While each piece of data might suggest some aspect of quantum critical behavior, the discussion appears disjoint to me and it is hard to connect all of the pieces because the same sets of data are not presented for all samples. Assuming that disorder is used as the tuning parameter of the purported quantum phase transition, the authors

might try to tie the pieces together by creating a kind of phase diagram directly comparing quantities such as scattering rate (see point 5) or temperature exponent or some other relevant quantity. In addition, it would be helpful if the authors could provide the temperature dependence of the resistivity for all samples over a typical range of say 300K to lowest temperatures measured, or at least for LNO_0811 where the MR is shown from 80K to 2K.

(5) One might expect near a quantum critical point that the scattering rate diverges or experiences a crossover, which could be obtained the coefficient A in the expression $r = r_0 + A \cdot T^N$. It would help corroborate some of the authors' claims if they should the behavior of this quantity as a function of disorder.

(6) The statement is made in the main text: "On samples showing large negative LMR, such as LNO_0811, its magnitude does not depend on the direction of the magnetic field (Supplementary Information), which cannot be explained by a weak localization mechanism." An isotropic negative MR was found earlier by LNO films due to weak localization, and was explained by a spin scattering contribution (see Ref. 27 in main text). It would appear rather that the system remains metallic in the negative MR regime, which would preclude weak localization. The authors might consider revising this argument.

(7) Fig. 4(d), it is known from (Nd,La)NiO₃ alloying experiments that LNO sits somewhat away from the AFM crossover regime in both bulk and thin films. The phase diagram proposed in Fig 4(d) seems to contradict this observation. Can the authors explain this?

(8) Based on the description of the localized Ni moments and conduction electrons, it would appear that both RKKY and Kondo type interactions might be relevant depending on the degree of disorder. Especially in the case laid out where each d8L site is modeled as a localized d8, the RKKY interaction may be dominant. How does this change the conclusions that the authors make?

(9) Relatively minor note: the labeling of the samples is quite confusing, as it is difficult to keep track of which are more or less disordered. I would recommend the authors change the nomenclature and label them not by the sample growth number, but rather by impurity level or even RRR.

(10) A relevant paper which corroborates the proximity to a QCP is based on tunneling spectroscopy of LNO, which indicates the presence of a "pseudogap." (doi.org/10.1063/1.4907771). The authors might want to cite this to support their claims.

Overall, it is a well-written paper that is certain to attract interest and make a significant contribution to the understanding of nickelate physics. My criticism is mainly based on a somewhat incomplete or unclear discussion. If the authors can address the major concerns, I would be certain to recommend publication.

Reviewer #2 (Remarks to the Author):

In their manuscript entitled "observation of an antiferromagnetic quantum critical point in high-mobility LaNiO₃", Liu and coworkers report on a series of transport and magnetotransport measurement and analysis on lanthanum nickelate thin films. They observe a regime where the resistivity varies linearly with temperature, often an indicator of proximity to a quantum critical point. Their claim is that this QCP is antiferromagnetic in nature.

The work is both important and timely as recent reports of a (infinite layer) nickelate seem to bring up some questions as to the role that magnetism plays in the superconductivity. The possibility of magnetic correlations in LaNiO_3 , specifically, seems to come to the surface every so often and clearly this study is relevant in the context of the appearance of single crystal LaNiO_3 in the last two to three years.

With regards the manuscript itself, to my knowledge, the measurements and analysis have been performed in a standard fashion. The speculation of an antiferromagnetic QCP also seems reasonable enough. I do, however, have some concerns that I will list below.

Major concern:

Throughout the manuscript the authors refer to their low residual resistivity samples as “clean” and, therefore, ascribe their observations to a level of purity in their LaNiO_3 that has not been achieved before. I have several comments/questions on the samples.

- I did think it was a neat idea to perform a systematic study of transport properties with varying cation ratio. This would, in some way, vary the impurity levels. However, this study is not systematic. The most conducting films were grown at a higher temperature, in a higher ozone environment and to a greater thickness than the less conducting films. Do the authors believe that the only relevant parameter across the series is just the La:Ni ratio?
- It is not clear to me what exactly the authors are suggesting is the source of the impurity. Do they believe that their low resistivity samples are simply the closest to the ideal 1:1 La:Ni ratio that any sample grower has reached? If so then I find it strange that the panels in Figure 3 are arranged from highest residual to lowest residual as the corresponding La:Ni goes 1.01 – 0.998 – 1.01 – 0.997 – 1. So, with the exception of the final panel there appears to be no correlation to the cation stoichiometry.
- The nickelates, very generally, are sensitive to the oxygen stoichiometry. In the extreme case, oxygen vacancies have been shown to entirely suppress the metal-insulator transition that the rest of the nickelate family is known for. That being said, in this manuscript, oxygen off-stoichiometry doping is not discussed at all. It really ought to be addressed as at least a possibility.
- The title of the manuscript mentions that the samples are “high-mobility” but there is no mention of mobility in the manuscript outside of the introduction and conclusion, where it is simply employed as a descriptive adjective. One sample, shown in the supplementary information, has had its Hall coefficient extracted but no calculation of the carrier mobility was shown. It would have been interesting to examine the relationship between the mobility and the signatures of quantum criticality.
- According to the supplementary information, the authors performed Rutherford Backscattering on one of their samples, the data from which they then used to calibrate the effusion rates. If they have access to such a technique why not measure the actual La:Ni ratio for all the samples? How reliable is the nominal cation ratio? If RBS is not convenient then why not EDX? Even x-ray diffraction can give some qualitative information on the off-stoichiometries.

Minor concerns:

- The authors ought to cite the work of their colleagues, Bi-Xia Wang et al, who published “antiferromagnetic defect structure in $\text{LaNiO}_{3-\delta}$ single crystals” in Physical Review Materials last year. In that paper they state “our results do confirm that AFM ordering is induced concomitant with ordered oxygen defect structures”. Given this result, is it likely that the x-axis of Figure 4d) corresponds to the oxygen content? Again, as I mentioned above, oxygen content is barely mentioned throughout.
- The statement in the second paragraph “it has long been suspected that the properties of LNO are influenced by its proximity to magnetic and structural instabilities, perhaps even by quantum fluctuations resulting from these instabilities” warrants some citations.
- It is generally accepted that the electronic ground-state of LaNiO_3 is a superposition of $3d^7$ and $3d^8$ so the $S = 1$ moment would be the extreme case of full

hybridisation, something that is unlikely.

- Figure 3 should have the corresponding La:Ni ratios displayed on each panel as this is the context in which this figure is introduced in the text.
- The supplementary information begins by referencing a 2003 work on MBE-grown LaNiO_3 but would it not be interesting to speculate on why the current samples might be so different to those?
- Table S1 is not relevant without adding columns detailing 1) upon which substrate the LaNiO_3 had been grown, 2) to what thickness the film was grown and 3) the growth technique and conditions. The authors should know how widely the resistivity can vary with these parameters.
- Previous studies on the substrate effect on LaNiO_3 transport properties find that LaAlO_3 is the best substrate for maximising conductivity. Did the authors try to grow on this substrate?
- Do the authors have surface-sensitive data, such as a topography from a scanning probe microscopy technique that could attest to the quality of their samples?

If the authors can adequately address my concerns, particularly regarding the purity or lack thereof, then I do think their work is worthy of publication. It should be interesting to those working in the field of nickelates, that, as I mentioned, has recently been boosted, and therefore well-suited for Nature Communications.

Reviewer #1 (Remarks to the Author):

The major claim of the submitted manuscript is that LaNiO₃ is proximal to an antiferromagnetic quantum critical point. The evidence to support the conclusion consists of: (1) a linear temperature dependence of the resistivity observed over a certain range of temperatures in various samples; (2) the crossover to quadratic temperature dependence with magnetic field; (3) the scaling of the magnetoresistance with some function of B/T (magnetic field/temperature); (4) the agreement of the temperature evolution of the temperature exponent of the resistivity with a theoretical model for scattering due to quantum spin fluctuations.

Given recent debate about the nature of the intrinsic ground state in LaNiO₃ and in the context of recently reported superconductivity in nickelate structures, the paper addresses an important, timely, and relevant topic. The presented data are certainly suggestive of quantum critical behavior and the phenomenology resembles other systems that are known to be influenced by antiferromagnetic quantum fluctuations such as heavy fermion compounds. However, I find several technical issues with the discussion and have a few pertinent questions. Therefore, I might recommend the manuscript for publication in Nature Communications, but I think the authors should address these issues before proceeding. They are enumerated below:

(1) A key part of the discussion is related to the influence of disorder on the quantum critical signatures of LaNiO₃. The relative amount of disorder is presumably deduced from the RRR of different samples, and it is speculated that the disorder is introduced from off-stoichiometry, leaving isolated Ni impurities. But, from the sample table shown in the supplement, there seems to be little correlation between the La/Ni ratio of the film and the RRR. Rather, the RRR seems to be more correlated with differences in growth conditions such as growth temperature and ozone nozzle distance, which may point to differences in oxygen incorporation in the films. One may then suspect that oxygen vacancies contribute the main impurities. This will still likely lead to localized Ni impurities. In any case, the notion that the disorder was systematically varied by the growth conditions appears doubtful. The authors should comment on the form of disorder, which may not be particularly known, and what kind of impact this has on the results.

Response:

We thank Referee 1 for their careful reading of our work and their insightful questions and comments. We address each of the questions raised below:

1. The Referee raises an important question about the actual stoichiometry of our samples, how we know what it is, how we control this to yield higher conductivities, and about the nature of disorder in our samples. As the Referee points out, the La/Ni ratio stays close to 1 in several of our samples, so the optimization of sample growth goes beyond that. There are several parts to our answer, where we address each of these issues:

- a. *Controlling disorder by optimizing growth conditions:* In order to obtain the cleanest samples, we worked on reducing the amount of disorder by gradually improving the growth conditions on all fronts. The different samples we present were obtained during this process, and the conditions for each sample are summarized in Table S2. The sample with the highest RRR was grown such that (i) nominal La/Ni ratio of 1/1 (using shutter time for La and Ni as determined by Rutherford Backscattering Spectroscopy (RBS) and X-ray characterization). It was important to have minimal drift of sources during growth to obtain the ideal cation stoichiometry (details in (c) below). (ii) Highest effective ozone pressure, which was controlled by varying ozone pressure in the chamber as well as nozzle distance to substrate. (iii) A substrate temperature of 615 °C during growth, which we found to be the most optimal amongst conditions we tried. We believe this temperature is low enough for oxygen uptake by the film, while also being high enough to promote good crystallinity. (iv) Our lowest resistivities were obtained in relatively thicker samples - 80 unit cells thick - grown in atomic layer-by-layer fashion (LaO-NiO₂-LaO, etc.) as mentioned in the paper and in the Supplementary Information with NiO₂ termination.
- b. *Quantifying disorder:* We cannot quantify the amount of Ni²⁺ and oxygen vacancies in our sample at these low concentrations using standard measurements like RBS or other spectroscopic tools. However, as the Referee mentions, we used the value of RRR and residual resistivity ρ_0 to quantify 'disorder' in our samples, and these follow measurable trends that we can correlate to the growth conditions (*see also c below*)

Changes: To make this part clearer, we have added the following text on page 3 of the paper: "To examine the role of disorder on quantum critical behavior of LNO, we grew a series of samples under different ozone pressure, substrate temperatures, and with slight variance in La/Ni ratio (< 2%). The growth conditions of each sample are presented in Supplementary Table S2. The main form of impurities introduced during the growth are oxygen vacancies and Ni²⁺, which also act as local magnetic moments. The sample with the highest RRR was grown under the highest effective ozone pressure, a substrate temperature of 615 °C, and a La/Ni ratio very close to 1."

- c. *Cation Stoichiometry, Oxygen Vacancies, and Resistivity:* The Referee mentions that the La/Ni ratio in Table S2 does not seem to correlate with the measured RRR and ρ_0 . Table S2 in the Supplementary Information has a column for the 'nominal' La/Ni ratio. However, this ratio does not account for drift in the atomic flux from the sources, which also affects the actual stoichiometry of our films. When we account for drift, the extrapolated 'actual' La/Ni ratio does indeed show a correlation with the measured RRR and ρ_0 . It is clear that samples with the highest RRR also have the 'actual' ratio of La/Ni closest to 1. Furthermore, oxygen vacancies also provide extra scattering and give rise to Ni²⁺ sites in their neighborhood, reduced from Ni³⁺. Even in samples where the La/Ni ratio is nearly ideal, we have a marked increase in RRR and lowering of ρ_0 upon increasing the effective ozone pressure by moving our nozzle closer to the sample, which presumably removes some O vacancies.

Changes: An additional column has been added to Table S2 to account for the ‘actual’ La/Ni ratio accounting for drift in sources. Text on pg. 3 of the paper reflects this as well.

- d. *Nature of disorder:* The Referee asks us to comment on the form of disorder. We note that for off-stoichiometric samples, excess Ni (the case for most of our ‘bad’ samples) would promote Ni²⁺ sites through local formation of NiO. Excess La also promotes Ni²⁺ sites through local formation of La₂O₃ (which getters oxygen) or La₂NiO₄ (see F.W. Wrobel *et al.*, *Appl. Phys. Lett.* **110**, 041606 (2017), and references therein). As mentioned earlier, O vacancies also promote the formation of Ni²⁺ sites. Thus, while we do not have a clear understanding of the nature of disorder in our samples, we can guess that the formation of a small concentration of Ni²⁺ sites is very likely in samples that are slightly off stoichiometric. This is borne out by our magnetoresistance measurements, which indicate the presence of spin-flip scattering from localized moments, consistent with Ni²⁺ sites, in all our ‘disordered’ samples.

Changes: Text on page 3 of the main paper, mentioned under **b** above, addresses this question.

(2) While the range of temperature that shows purported linear behavior can be explained by increased disorder, it is not clear why the absolute temperature at which linear behavior arises should increase. What is the origin of this?

Response: According to Rosch’s model, $\rho(T)$ near an AFM QCP is determined by the interplay between isotropic scattering from impurities and scattering from AFM spin fluctuations which are most pronounced for ‘hot’ regions of the Fermi surface nested by a characteristic AFM wavevector Q . In very clean samples, the non-nested ‘cold’ regions short out the ‘hot’ regions yielding a T^2 dependence. As disorder is introduced, the carriers get scattered into the ‘hot’ regions and a $T^{3/2}$ behavior emerges at low T . The temperature at which this crossover begins to occur is higher for greater strength of disorder. The regime where the resistivity changes linearly with temperature, i.e. $\Delta\rho(T) \sim T$ is a crossover between the T^2 (or an exponent higher than $3/2$) regime at higher T and the $T^{3/2}$ regime at lower T . A bit more precisely, as a result of scattering, the fermion distribution function develops a width Δk around ‘hot’ regions connected by the AFM scattering wavevector Q . The change in resistivity $\Delta\rho$ depends on Δk , which is a function of temperature and impurity levels. In the crossover regime of interest, $t + r < (\Delta k)^2 < 1$, where t is the reduced temperature as described in the main text and r is the distance away from the quantum critical point, $\Delta\rho \approx t^{2/(5-d)} x^{(4-d)/(5-d)}$, in which d is the dimensionality and $x (<1)$ is a measure of the disorder level ($\sim 1/k_F l$). For a 3D system that is very near the AFM QCP ($r \sim 0$), $\Delta\rho \approx t\sqrt{x}$. The temperature range ($T_{\text{low}} < T < T_{\text{high}}$) in which this relation holds is $x < t < \sqrt{x}$. Therefore, as the disorder level (x) increases, the temperature for the onset of linear resistivity, where $\Delta\rho(T) \sim T$, also increases. However, the ratio $\frac{T_{\text{high}}}{T_{\text{low}}} = 1/\sqrt{x}$ shrinks – so the linear in T resistivity only holds over a relatively short range of T (on a log scale). Conversely, as disorder decreases, the ratio of temperature $\frac{T_{\text{high}}}{T_{\text{low}}}$ becomes larger, but T_{high} has a lower value. These trends are observed in our

data. These ideas are presented briefly in the main text and Supplementary Information and are detailed in the relevant references to the works of Rosch.

Fig. R1. Temperature dependence of resistivity and resistivity exponents for low-resistivity samples.

(3) Showing a line on top of a small region of temperature does not necessarily make a strong case for linear T behavior, and the claim that linear T behavior is seen over a decade is not obvious from the plot of Fig. 1(b), as the resistivity deviates from linear rather clearly starting around 0.7K or so. Can the authors show more definitive evidence for the linear T behavior? For example, the plot of the temperature exponent in Fig. 1(c) provides more valuable information. I

suspect that a similar plot for the other samples would help clarify which truly show linear T behavior (and over which range) as well as for which samples the linear T region is just a crossover between the super- and sub-linear regimes.

Response: Fig. 1c may indeed be misleading because it does not show a full decade of linear resistivity, a point we make early in the paper. We agree with the Referee's suggestion and have replaced the dataset with one that shows linear behavior down to lower temperatures over a full decade of temperature. Furthermore, our analysis goes beyond showing a linear fit. Shown in Fig. R1 are the temperature dependences of the resistivity and the exponent for several clean LNO samples. Figure R1 (a, c) are the raw resistivity data measured on two LNO_24 samples using a helium 3 cryostat (down to 270 mK) and a dilution refrigerator (down to 25 mK), respectively. The red solid line is a smooth interpolation of the data from which the resistivity exponent is obtained by taking the derivative. Shown in Fig. R1(b, d) are the temperature dependences of exponents. In both cases, the resistivity exponent drops from about 1.7 to the range of 1.0 ± 0.1 as indicated by the shaded rectangle. In particular, the exponent stays within 1.0 ± 0.1 from 0.1 K to 1.1 K, or approximately one decade of temperature, as seen in Fig. R1(d). In Fig. R1(b), we included the resistivity exponent of another sample, LNO_18, the resistivity of which is presented in the inset of Figure 2a of the main text.

Changes: To clarify this point, we have replaced the data in Figure 1b of the main text with the measurement results on LNO_24 using the dilution refrigerator. The corresponding exponent analysis is also updated in Figure 1c of the main text.

Fig. R2. Analysis of resistivity exponent for higher resistivity samples.

Higher resistivity samples: We have analyzed the resistivity exponents for LNO_3, LNO_4 and LNO_7, which are shown below in Fig. R2. It can be seen in the plot, the resistivity exponents are similar among these samples at high temperatures, while they differ appreciably at low temperatures. In the low temperature range ($T < 20$ K), the resistivity exponent of LNO_3 initially drops, crossing 1, and shows sublinear value (< 1) in a broad temperature range. The quasilinear region (shaded area) for LNO_3 is relatively narrow. For samples LNO_4 and LNO_7, the resistivity exponent stays in the quasilinear region over a wider temperature range. Regarding the linear ρ vs. T , we cannot draw a completely definitive conclusion of its nature. For clean samples, such as LNO_24, the transport behavior is more aligned with Rosch's model. In less clean samples, the scattering from Ni^{2+} magnetic moments produces significant sublinear resistivity behavior. The transport properties of such a system in the presence of both AFM spin fluctuations and magnetic impurities have not been fully considered in current theories.

(4) The discussion first begins with a clean sample which shows T linear behavior at low temperatures, then another one for which a high magnetic field restores T^2 behavior, then a group of samples of different disorder, then a disordered sample showing negative MR. While each piece of data might suggest some aspect of quantum critical behavior, the discussion appears disjoint to me and it is hard to connect all of the pieces because the same sets of data are not presented for all samples. Assuming that disorder is used as the tuning parameter of the purported quantum phase transition, the authors might try to tie the pieces together by creating a kind of phase diagram directly comparing quantities such as scattering rate (see point 5) or temperature exponent or some other relevant quantity. In addition, it would be helpful if the authors could provide the temperature dependence of the resistivity for all samples over a typical range of say 300K to lowest temperatures measured, or at least for LNO_3 where the MR is shown from 80K to 2K.

Response: In order to answer the Referee's question, we clarify here the logical flow of our work and make changes to the manuscript as needed.

The main findings in the paper are supported by three different transport measurements on our LNO samples, across several samples:

- i. The first one is the observation of a linear resistivity over a wide temperature range $0.1 \text{ K} < T < 1.1 \text{ K}$, which is only observed in clean samples with $\text{RRR} > 18$ as shown in Fig. 1 of the main text. For less clean samples (Figure 3 of the main text), quasilinear behaviors are seen only in a narrow temperature window and at elevated temperatures, consistent with Rosch's theory.
- ii. The second measurement (Fig. 2) is the restoration of Fermi-liquid behavior in our cleanest samples. A quadratic temperature dependence of resistivity is obtained by applying a magnetic field. This feature is also observed only in clean samples that show linear ρ vs. T in zero magnetic field. See additional answers in the next questions.
- iii. The third (Fig. 4) is the evolution of the magnetoresistance from positive (clean samples) to negative values, as a function of increasing disorder. For the disordered samples, we

observe that the negative magnetoresistance scales in a manner consistent with spin-flip scattering.

The parameter that captures the qualitative variations in our data is disorder or scattering rate as suggested by the Referee. However, a point that perhaps needs more emphasis is that disorder is known to lead to *deviations* from quantum critical behavior near AFM QCPs. As an example, in CeCoIn_5 , magnetic dopants can give rise to droplets of magnetism that obscure quantum critical behavior. It is in this spirit that we think about the effects of impurities de-tuning the system away from quantum criticality (S. Seo *et al.*, *Nature Physics* **10**, 120 (2013)) towards a phase with short-range AF order near the QCP.

In the phase diagram of Fig. 4, what we intend to show is that in the clean limit, we observe clear signatures of quantum critical behavior. When we introduce impurities, this leads to deviations from quantum critical behavior, presumably due to scattering of carriers from localized spins, and due to interactions between these localized spins that can lead to short-range magnetic order.

Regarding the full temperature dependence of the resistivity, we did measure them in the temperature range from 2 K to 300 K for sample LNO_3, LNO_4, LNO_5 and LNO_7. The data are presented in the Supplementary Information. We are attaching it here as Fig. R3 for reference.

Fig. R3 Temperature dependence of resistivity for different samples. The temperature range is 2 K to 300 K, and the x-axis is plotted as $T^{1.5}$.

Changes: To make the second point clear, a description “We note that, the crossover to LFL in the resistivity is seen only in samples with $\text{RRR} \geq 18$. Therefore, the presence of disorder in LNO can detune the system away from the QCP. ” is added to the main text. Also, we make clear that

the role of disorder is not as a tuning parameter in this phase diagram but rather something that clouds the intrinsic quantum critical nature of transport in LNO.

(5) One might expect near a quantum critical point that the scattering rate diverges or experiences a crossover, which could be obtained the coefficient A in the expression $r = r_0 + A \cdot T^N$. It would help corroborate some of the authors' claims if they should the behavior of this quantity as a function of disorder.

Response: As we described in the previous question, the field-induced quadratic temperature dependence of the resistivity is observed only in clean samples. The resistivity crosses over from linear to $\rho = \rho_0 + AT^2$, with A equal to 0.008 and $0.007 \mu\Omega \text{ cm K}^{-2}$ for $B = 9 \text{ T}$ and 18 T , respectively. These values of A are about 3 to 4 times larger than those, around $0.002 \mu\Omega \text{ cm K}^{-2}$, reported in the literature for LaNiO_3 . The large value of the coefficient A observed here bears characteristics of enhanced scattering near the QCP, however, more detailed magnetic field dependence measurements are necessary in order to draw a quantitative conclusion on this aspect.

For samples with increased disorder, while we do not observe the restoration of Fermi-liquid behavior (up to 9 Tesla), the sublinear part of the resistivity curve gets suppressed and correspondingly the resistivity exponent increases to some value around 1.5. See, for example, Fig. S8a.

Changes: To strengthen the QCP part, we have added the following description into the main text: "The A coefficients are 8×10^{-3} and $7 \times 10^{-3} \mu\Omega \text{ cm K}^{-2}$ for $B = 9 \text{ T}$ and 18 T , respectively. These values of A are about 3 to 4 times larger than those, $\sim 2 \times 10^{-3} \mu\Omega \text{ cm K}^{-2}$, reported for LNO previously [Son2010, Zhang2017, Guo2018], which is a suggestive of enhanced scattering near the QCP. "

(6) The statement is made in the main text: "On samples showing large negative LMR, such as LNO_0811, its magnitude does not depend on the direction of the magnetic field (Supplementary Information), which cannot be explained by a weak localization mechanism." An isotropic negative MR was found earlier by LNO films due to weak localization, and was explained by a spin scattering contribution (see Ref. 27 in main text). It would appear rather that the system remains metallic in the negative MR regime, which would preclude weak localization. The authors might consider revising this argument.

Response: Weak localization by itself cannot explain our data – in our cleanest samples, we see positive MR, and no upturn in resistivity. In disordered samples, we observe a negative isotropic MR, which is inconsistent with weak localization in thin films. We note that in Ref. 27 of our prior submission (Scherwitzl *et al.*, PRL 2011), the authors invoked an additional spin fluctuation term (somewhat like spin-flip scattering) on top of weak localization effects to arrive at a nearly isotropic negative MR. We observe B/T scaling of the negative LMR in our experiment consistent with single-impurity spin-flip scattering – this part has been missed in prior studies. Thus, we conclude that weak localization is not at play, and the negative MR is due to spin-flip scattering from impurities.

Changes: We have taken the Referee's suggestion and revised the argument in the main text as: "In samples showing large negative LMR, such as LNO_3, its magnitude does not depend on the direction of the magnetic field (see Supplementary Fig. S9). The resistivity also shows metallic behavior down to the lowest temperature. These cannot be explained by a weak localization mechanism [Scherwitzl2011]."

(7) Fig. 4(d), it is known from (Nd,La)NiO₃ alloying experiments that LNO sits somewhat away from the AFM crossover regime in both bulk and thin films. The phase diagram proposed in Fig 4(d) seems to contradict this observation. Can the authors explain this?

Response: Indeed, in alloying (La_{1-x}Nd_x) on the A-site in LNO, it takes substantial values of x (> 0.5) to obtain long range AFM order (J. Blasco, J. Garcia, *J. Phys. Cond. Matter.* **6**, 10759 (1994)). One possible reason is that alloyed samples have large amount of disorder which can drive the system away from the critical point. On the other hand, LNO does seem to be very close to an AFM instability, particularly when considering the effects of O-vacancies. In fact, in bulk single crystals, it has been shown that for only ~ 2.6 % concentration of O-vacancies (LaNiO_{3- δ} , where $\delta = 0.079$), LNO adopts long-range AFM ordering at the $\frac{1}{4}, \frac{1}{4}, \frac{1}{4}$ wavevector (B. Xia *et al.*, *Phys. Rev. Materials* **2**, 064404 (2018)). However, we cannot determine the exact position where LNO stands relative to the critical point, therefore we used a band with a finite width to represent LNO in the phase diagram.

Changes: In previous version, the rectangle representing LNO sits in the middle of the crossover regime. That may be misleading in that it suggests that there is a long range order in pure LNO. We have drawn the band to the right side of the critical point in the revised Fig. 4d.

(8) Based on the description of the localized Ni moments and conduction electrons, it would appear that both RKKY and Kondo type interactions might be relevant depending on the degree of disorder. Especially in the case laid out where each d8L site is modeled as a localized d8, the RKKY interaction may be dominant. How does this change the conclusions that the authors make?

Response: The Referee's point is correct. The RKKY interaction between localized magnetic moments via conduction electrons may be present. Here, we find that introducing a small concentration of Ni²⁺ sites will first introduce new scattering mechanisms (as evidenced in our data); at higher concentrations it presumably leads to short-ranged AFM order which might be a consequence of the RKKY interaction. The scattering will obscure the quantum critical fluctuations and the short-ranged AFM order will detune the system away from the QCP, as we have seen from the sublinear resistivity data. These behaviors do not change our main conclusion, because in Rosch's model, non-universal behavior can occur around the critical point in the low T limit due to things, such as the magnetic impurities seen here, missing in the simple theory.

(9) Relatively minor note: the labeling of the samples is quite confusing, as it is difficult to keep track of which are more or less disordered. I would recommend the authors change the nomenclature and label them not by the sample growth number, but rather by impurity level or even RRR.

Changes: All of the labelings of the sample have been changed based on the sample's RRR.

(10) A relevant paper which corroborates the proximity to a QCP is based on tunneling spectroscopy of LNO, which indicates the presence of a "pseudogap." (doi.org/10.1063/1.4907771). The authors might want to cite this to support their claims.

Changes: We have added the reference to the main text.

Overall, it is a well-written paper that is certain to attract interest and make a significant contribution to the understanding of nickelate physics. My criticism is mainly based on a somewhat incomplete or unclear discussion. If the authors can address the major concerns, I would be certain to recommend publication.

We thank the Referee for the positive recommendation.

Reviewer #2 (Remarks to the Author):

In their manuscript entitled "observation of an antiferromagnetic quantum critical point in high-mobility LaNiO_3 ", Liu and coworkers report on a series of transport and magnetotransport measurement and analysis on lanthanum nickelate thin films. They observe a regime where the resistivity varies linearly with temperature, often an indicator of proximity to a quantum critical point. Their claim is that this QCP is antiferromagnetic in nature.

The work is both important and timely as recent reports of a (infinite layer) nickelate seem to bring up some questions as to the role that magnetism plays in the superconductivity. The possibility of magnetic correlations in LaNiO_3 , specifically, seems to come to the surface every so often and clearly this study is relevant in the context of the appearance of single crystal LaNiO_3 in the last two to three years.

With regards the manuscript itself, to my knowledge, the measurements and analysis have been performed in a standard fashion. The speculation of an antiferromagnetic QCP also seems reasonable enough. I do, however, have some concerns that I will list below.

Major concern:

Throughout the manuscript the authors refer to their low residual resistivity samples as "clean" and, therefore, ascribe their observations to a level of purity in their LaNiO_3 that has not been achieved before. I have several comments/questions on the samples.

- I did think it was a neat idea to perform a systematic study of transport properties with varying cation ratio. This would, in some way, vary the impurity levels. However, this study is not systematic. The most conducting films were grown at a higher temperature, in a higher ozone

environment and to a greater thickness than the less conducting films. Do the authors believe that the only relevant parameter across the series is just the La:Ni ratio?

Response: The Referee's point is correct - the La/Ni ratio is not the only parameter that controls the disorder level in these LNO films. In most of our samples presented here, we in fact tried to keep the La/Ni ratio at the optimal level. We refer the Referee to our response to the first comment of Referee 1, where we detail how we control other growth parameters such as the ozone pressure, growth temperature, sample thickness and termination layer, all of which play a role in determining the transport behavior of the sample.

Changes: To clarify this aspect, we have revised the description of sample growth in the main text. See also the second point in the summary of changes.

- It is not clear to me what exactly the authors are suggesting is the source of the impurity. Do they believe that their low resistivity samples are simply the closest to the ideal 1:1 La:Ni ratio that any sample grower has reached? If so then I find it strange that the panels in Figure 3 are arranged from highest residual to lowest residual as the corresponding La:Ni goes 1.01 – 0.998 – 1.01 – 0.997 – 1. So, with the exception of the final panel there appears to be no correlation to the cation stoichiometry.

Response: The Referee is correct. The La/Ni ratio is not the only parameter controlling the sample's final resistivity. We have addressed this part together with the previous comment when we addressed the concerns of Referee 1, and made changes to the main text and Supplementary Information clarifying our growth conditions.

- The nickelates, very generally, are sensitive to the oxygen stoichiometry. In the extreme case, oxygen vacancies have been shown to entirely suppress the metal-insulator transition that the rest of the nickelate family is known for. That being said, in this manuscript, oxygen off-stoichiometry doping is not discussed at all. It really ought to be addressed as at least a possibility.

Response: The oxygen stoichiometry was indeed very important, and the nickelates are known to be susceptible to oxygen deficiency. In most of our growths, we aimed to have ideal La/Ni stoichiometry. However, once this was optimized, we realized that it was possible for us to grow very high-RRR samples by increasing the effective ozone pressure by reducing the distance between the ozone delivery nozzle and the substrate.

Changes: To address this and the previous two comments, we have revised the section of sample growth in the main text which is copied below:

“To examine the role of disorder on quantum critical behavior of LNO, we grew a series of samples under different ozone pressure, substrate temperatures, and with slight variance in La/Ni ratio (< 2%). The growth conditions of each sample are presented in Supplementary Table S2. The main form of impurities introduced during the growth are oxygen vacancies and Ni²⁺, which also act as local magnetic moments. The sample with the highest RRR was grown under the highest effective ozone pressure, a substrate temperature of 615 °C, and a La/Ni ratio very close to 1.”

- The title of the manuscript mentions that the samples are “high-mobility” but there is no mention of mobility in the manuscript outside of the introduction and conclusion, where it is simply

employed as a descriptive adjective. One sample, shown in the supplementary information, has had its Hall coefficient extracted but no calculation of the carrier mobility was shown. It would have been interesting to examine the relationship between the mobility and the signatures of quantum criticality.

Response: The Referee correctly points out that the mobility itself was not used widely in this study. We used this term to emphasize the high purity & conductivity of the samples (which we have characterized using the residual resistivity and RRR). For reference, the mobility on the cleanest sample, assuming a single band and isotropic scattering, is about $160 \text{ cm}^2 \text{ V}^{-1} \text{ s}^{-1}$ at $T = 2 \text{ K}$. The mobilities of our samples are largely proportional to their conductivities or RRR, which we have been using in our study, and quantum critical behavior is observed in samples with high mobility.

Change: We have added the mobility value of this sample to the main text.

- According to the supplementary information, the authors performed Rutherford Backscattering on one of their samples, the data from which they then used to calibrate the effusion rates. If they have access to such a technique why not measure the actual La:Ni ratio for all the samples? How reliable is the nominal cation ratio? If RBS is not convenient then why not EDX? Even x-ray diffraction can give some qualitative information on the off-stoichiometries.

Response: For the RBS measurements, we need to avoid the background signal from elements in the substrate in order to get an accurate chemical composition for the film. Since the LNO samples were grown on LSAT, which contains La, there is a large overlap with the signal from the LNO film, making the determination of the stoichiometry using RBS not viable. For calibration, we grew samples on MgO substrates using exactly the same growth parameters (temperature, ozone pressure) specifically for the RBS measurement. A good RBS calibration gives us the value of the La/Ni ratio with an error of 1% or less. However, the absolute values of the La and Ni content is known only approximately to within 10 - 15% accuracy. In order to calibrate the exact amount of La and Ni in the film, we used X-ray diffraction and reflectivity to measure the lattice parameter and film thickness of an LNO film that we grew on our LSAT substrates. This information allows a calibration of our La and Ni fluxes to be better than 1%. The deposition rate for both La and Ni were typically very stable, where the drifts could be less than 0.3%/h, which allows for control of our compositions to ~ 1% level or better when the drifts are low.

We tried using EDX, however, it also picked up a significant amount of signal from the substrate. Also, when we perform the EDX measurement on samples grown on MgO, we yielded an accuracy of only $\pm 5\%$, which was used as a check.

Minor concerns:

- The authors ought to cite the work of their colleagues, Bi-Xia Wang et al, who published “antiferromagnetic defect structure in $\text{LaNiO}_{3.5}$ single crystals” in Physical Review Materials last year. In that paper they state “our results do confirm that AFM ordering is induced concomitant with ordered oxygen defect structures”. Given this result, is it likely that the x-axis of Figure 4d) corresponds to the oxygen content? Again, as I mentioned above, oxygen content is barely mentioned throughout.

Changes: We have added the reference for Bi-Xia Wang *et al.*'s work in the context of discussing the possibility of short-range AFM order.

- The statement in the second paragraph “it has long been suspected that the properties of LNO are influenced by its proximity to magnetic and structural instabilities, perhaps even by quantum fluctuations resulting from these instabilities” warrants some citations.

Changes: We have added four references (Hoffman *et al.*, *Phys. Rev. X* **6** (2016), Fabbris *et al.*, *Phys. Rev. B* **98** (2018), S. Allen *et al.*, *APL Mater.* **3** 6 (2015) and A. Subedi, *SciPost Phys.* **5** (2018)) in that context.

- It is generally accepted that the electronic ground-state of LaNiO₃ is a superposition of 3d⁷ and 3d⁸L so the S = 1 moment would be the extreme case of full hybridisation, something that is unlikely.

Response: For LNO, the ground state is indeed hybridized as the Referee suggests, with 3d⁷ having S = 1/2 on each Ni site, and 3d⁸L with the Ni S = 1 spin screened by the anti-aligned ligand hole spin to make it also effectively S = 1/2. On the other hand, when there are oxygen vacancies, the ligand hole is removed and a bare Ni²⁺ site with S = 1 is obtained.

In the low temperature insulating state seen in RNiO₃, it is now believed that a bond disproportionation with alternating d⁸L² and d⁸L⁰ sites along [111] – with alternating S = 0 and S = 1, spins respectively - is a likely description. See S. Catalano *et al.*, *Rep. Prog. Phys.* **81**, 046501 (2018).

- Figure 3 should have the corresponding La:Ni ratios displayed on each panel as this is the context in which this figure is introduced in the text.

Response: The disorder in our samples is characterized by the residual resistivity ratio. The La:Ni ratios are nearly identical in some cases, and varying the ozone pressure had the strongest impact. In all cases, reduced residual resistivity and increased residual resistivity ratio (RRR) resulted in an enhanced range in temperature where the resistivity shows a linear temperature dependence.

Changes: We have revised the sample labeling by using RRR throughout manuscript.

- The supplementary information begins by referencing a 2003 work on MBE-grown LaNiO₃ but would it not be interesting to speculate on why the current samples might be so different to those?

Response: We can only speak for what we did to improve our sample quality. In fact, in earlier work our ‘good’ samples typically had a residual resistivity $\rho_0 \sim 30 - 40 \mu\Omega \text{ cm}$, comparable to the work from 2003 from the Goldman group in Minnesota. Our layer-by-layer growth approach is in fact exactly the same as theirs. To improve our ρ_0 values further, we did the following:

1. An increased effective ozone pressure. The delivery of the ozone in our setup is through a tube that is cooled by chilled water over its entire length – this prevents the dissociation of O₃ into O₂ in the tube during delivery. The ozone delivery tube is also mounted on a linear motion feedthrough with a bellows seal, so we were able to vary the distance between the end of the tube and the substrate by as much as a factor of 2. This raises the

effective flux of ozone at the substrate by more than a factor 2 (though not quite a factor of 4 as there is a collimating effect of the tube).

2. Anneal the growing film in ozone for 30 seconds after each NiO₂ layer – the RHEED pattern gets sharper during this anneal.
3. Fine tune La to Ni ratio, using very stable effusion cell fluxes whose rates drifted only minimally during growth.
4. Appropriate growth temperature. Our hypothesis is that there is an optimal temperature range that is high enough to promote good crystallinity and low enough to prevent oxygen loss from the film during growth.

• Table S1 is not relevant without adding columns detailing 1) upon which substrate the LaNiO₃ had been grown, 2) to what thickness the film was grown and 3) the growth technique and conditions. The authors should know how widely the resistivity can vary with these parameters.

Changes: We have revised Supplementary Table S1 to include information about the substrates used, thickness of the films and relevant growth techniques.

• Previous studies on the substrate effect on LaNiO₃ transport properties find that LaAlO₃ is the best substrate for maximizing conductivity. Did the authors try to grow on this substrate?

Response: The Referee is correct, in that LaAlO₃ has a slightly smaller lattice constant than LSAT. This will impose slight compressive strain in the LNO film, which could enhance the conduction band width. However, the LaAlO₃ substrate typically has a large mosaic spread. Therefore, we avoided using LaAlO₃ for the growth of LNO as we thought it might get in the way of detailed X-ray studies.

- Do the authors have surface-sensitive data, such as a topography from a scanning probe microscopy technique that could attest to the quality of their samples?

Fig. R4 AFM scan (top) and XRR measurement (bottom) on sample LNO_3.

Response: We did AFM measurements on only some of our samples. With our limited data, we found that the surface roughness does not correlate well with the sample's conductivity. Shown in Fig. R4 is the AFM measurement on sample LNO_3, which has low RRR compared to other samples. The RMS roughness is about 0.1 nm from the AFM measurement. The X-ray reflectivity measurement on the same sample shows a roughness of about 0.38 nm. For the high-mobility sample, such as LNO_18, we took only the X-ray reflectivity measurement. Shown in Fig. R5 is the analysis of the X-ray reflectivity measurement which gives a surface roughness of about 0.86 nm, higher than that of LNO_3. However, the RHEED pattern of those high mobility samples shows good surface crystallinity (see Fig. S2 in the Supplementary Information).

Fig. R5 X-ray reflectivity measurement on sample LNO_18. The roughness is about 0.86 nm.

If the authors can adequately address my concerns, particularly regarding the purity or lack thereof, then I do think their work is worthy of publication. It should be interesting to those working in the field of nickelates, that, as I mentioned, has recently been boosted, and therefore well-suited for Nature Communications.

We thank the Referee for their assessment.

Reviewer #3

In this manuscript, Bhattacharya and co-workers studied the quantum critical properties of high mobility metallic LaNiO₃ and experimentally tried to demonstrate the existence of antiferromagnetic critical point in this system. Their conclusion relies on the observation of T-linear resistivity below a certain temperature, which is often attributed to quantum critical fluctuations, as in curates and pnictides. In addition, they also observed sublinear scaling of resistivity below a certain temperature, which can arise from both impurities and short-range antiferromagnetic orders. By applying external magnetic field such sublinear T dependence can be suppressed which indicates that it possibly arises from short-range antiferromagnet order. However, at the same time the T-linear dependence gets replaced by a T² dependence. While these observations are claimed to consistent with the ones made in other heavy-fermion compounds, overall I find the evidence of a genuine AFM quantum critical point or quantum critical fluctuations to be somewhat inconclusive. More experimental works need to done in this direction to pin down the actual source of such anomalous transport behavior. Nonetheless, this is the first experimental work in metallic nicketes suggesting a possible existence of critical point therein. Hence, this paper can be published (purely on the merit of the experimental effort, as the theoretical work essentially recapitulates some existing ideas) in Nature Communication.

However, prior to publication of this article the authors should address the following issues and make appropriate adjustments in the manuscript. The current version of the manuscript definitely requires more work before publication.

We thank the Referee for their careful reading and overall positive recommendation. We address all the additional points raised, as outlined below.

1. Authors should tone down the claim of AFM critical point a bit, as the range of T -linear behavior is quite small, and the exponent α is not fixed to unity, rather it is fluctuating.

Response: We have done an analysis on the resistivity exponent on samples measured using a dilution refrigerator. The value of the exponent falls into a window of 1.0 ± 0.1 , for temperatures from 0.1 K to 1.1 K, a range of about one order of magnitude. The exponent does vary at higher temperatures, however, according to Rosch's scenario, this is to be expected – the variation of our resistivity exponent as temperature decreases shows trends similar to those predicted in Rosch's theory. The T -linear regime is a crossover between high- T and low- T behaviors, though it can extend for a very substantial range of temperature (\sim one order of magnitude in our cleanest samples). See also Fig. R1 in our response to Referee 1.

2. Why do authors commit to Herz-Millis-Moriya formalism in an itinerant system? when we know quite surely that this theory does not work in metallic systems, where gapless fermionic degrees of freedom exists near the critical point.

Response: Actually, Hertz-Millis-Moriya theory was designed to describe itinerant metallic systems. We agree with the Referee that the theory is incomplete, in that one treats the spin fluctuations and fermions as separate entities, which is an approximation (that is, the theory is obtained by integrating out the fermions, which is an approximation when there is a Fermi surface). A more correct theory could lead to different critical exponents, but as Rosch points out, all of this is modified in the presence of disorder. The main point that we take away from Rosch's theory is that the linear ρ vs T is a crossover between two regimes in the clean limit, an exponent higher than $3/2$ at high temperature, to something closer to $3/2$ at very low temperature – which gives rise to a quasi-linear in T dependence in the crossover region. We also note that Rosch's theory is the only one we know that deals with the effects of disorder on transport properties near an AFM QCP.

3. Authors should do more measurements on magneto transport, and show the scaling of resistivity with both T and B . Note that there are two possible scenarios that can arise in the scaling of $\rho(T, B)$. It can either be (1) $\rho(T, B) = aT + bB$, where a and b are two unknown (and non-universal) coefficients, as been observed in cuprates, or (2) $\rho(T, B) = \sqrt{aT^2 + bB^2}$ as been observed in pnictides. I strongly believe that authors have the data to extract such scaling of magneto resistance. This scaling will at least provide more conclusive evidence in favor (or against) of an AFM critical point, instead of just relying on the T -linear scaling of resistivity, which can only be observed over a narrow temperature window.

Fig. R6. Linear positive magnetoresistance observed on clean samples. The red dotted line is a guide to the eye.

Response: We performed magnetoresistance measurements on a high mobility sample in fields up to 18 T. We found that the positive MR on high-RRR samples becomes linear at fields above about 5 T, which is shown in Fig. R6. The slope of the linear MR is $0.027 \mu\Omega \text{ cm T}^{-1}$. In comparison, the slope of the linear ρ vs. T curve for the same sample is $0.015 \mu\Omega \text{ cm K}^{-1}$. We had considered both equations provided in the Referee's comments, and we found that the first equation would phenomenologically fit what we observe, while the second equation would produce too small of a temperature dependence of MR. The similarity in energy scales between the slope of the linear ρ vs. T and the MR may support the existence of a quantum criticality in LNO. However, we did not pursue the scaling analysis on LNO for two reasons:

1. The linear ρ vs. T observed in LNO may have a different origin than those observed in cuprates. As discussed in the main text (Fig 1c) and Supplementary Information, the linear

ρ vs. T observed here comes from the interplay of AFM spin fluctuations and impurity scatterings. It is a crossover between the T^2 (or an exponent higher than 3/2) regime at higher T and the $T^{3/2}$ regime at lower T . See also our response to the second comment of Referee 1.

2. The magnitude of the positive MR is sensitive to the disorder level, which we consider to be an extrinsic factor. One of the major sources of disorder in LNO comes from Ni^{2+} sites, which are also magnetic. Because of the spin-flip scatterings from local magnetic moments, the magnitude of the positive MR decreases and can become even negative in samples with elevated disorder level, see Fig. 4 of the main text.

4. Typically in metallic systems, AFM or SDW critical point gets masked by superconductivity at lowest temperature. I am bit surprised that author did not make any comment on this. This issue needs to be addressed.

Response: We do not observe superconductivity at the lowest temperature (we checked). This type of behavior is not unprecedented. Particularly, for antiferromagnetic chromium, there is also no superconductivity at the quantum critical point. Both LNO and Cr are 3D metals, and in fact ARPES data on LNO indicate that it has a Fermi surface similar to Cr.

Reviewers' comments:

Reviewer #1 (Remarks to the Author):

The authors have provided a comprehensive response to my comments as well as those of Reviewers #2 and 3. I am impressed by the detail of the answers and the changes made to the manuscript/supplement. I still have some slight misgivings regarding some aspects of the interpretation/presentation, which really ought to be addressed before proceeding. But, I think the manuscript as a whole is significantly improved and I would recommend its publication. The experimental results and the timeliness of the paper will certainly interest the community.

One thing I would insist before proceeding is that the authors include some form of the Figs. R1 and R2 shared with the referees in the supplement and/or main text. I think it is quite necessary for a reader to examine in detail the resistivity exponents for different samples, as this is the major important claim of the paper. I encourage the authors to include it in the main text, but if not there, then it should at the very least be in the supplement.

In addition, I encourage the authors to consider my comments below, which I think would improve the manuscript.

- To address a point of Reviewer #2, maybe the authors should change the title to say "high-purity LaNiO₃ films" which is probably more precise in the context of this paper. Alternatively, where the authors state, "This sample has a ...mobility of about 160 cm²/Vs at T = 2 K. The corresponding RRR is about 24. This is so far the highest RRR reported for LNO" if they could compare their mobility to the literature and say this is the highest mobility reported (at the same temperature), then the title of the paper would make sense.

- Based on the data shown in Figs. R1 and R2 and the main text, I am still not convinced of the statement that "the resistivity $\rho(T)$ shows a linear temperature dependence over a decade of T below ~ 1.1 K in our cleanest samples." The authors should temper their claim a bit and write "almost a decade in T" which would be a more accurate representation of the data.

- It would benefit readers if the authors would comment on/rule out other mechanisms for linear T resistivity besides the quantum critical AFM scenario. They briefly mention "Phonons are nominally not relevant in this low temperature regime given that the Debye temperature of LNO is above 400 K." But, a sentence or two about other potential origins such as charge fluctuations, Umklapp scattering or diffusive transport would be useful.

- The authors have now written in the manuscript on page 2: "The disorder level in the sample may be characterized by the residual resistivity or RRR of the sample, assuming that the Matthiessen rule holds." However, in Rosch's theory, upon which much of the experimental conclusions are based, Matthiessen's rule is expressly broken in the crossover regime where the interesting temperature dependences are found. While this may not invalidate the claim that the RRR is representative of the level of disorder, the authors should be careful in their arguments and at least revise the aforementioned statement.

- Given the importance of the connection, some mentions might be made about superconductivity – both the relation of the quantum critical point to superconductivity (as pointed out by Referee #3) and with reference to the observed superconductivity infinite layer nickelates.

Reviewer #2 (Remarks to the Author):

It was very much my impression (and the impression of at least one other reviewer judging by their comments) in the initially-submitted manuscript that the authors were claiming a systematic control of disorder by cation ratio. As this has now been clarified and the possibility of oxygen vacancies has been mentioned I feel that this issue, which was my main one, has been resolved.

However, two of my minor concerns have not been addressed satisfactorily. If these modifications are made then I believe the paper will be sufficiently complete for publication.

1. Perhaps my point concerning the electronic configuration was not clear enough. The hybridisation will never be complete and the Ni-O bond will always have a mixed covalent/ionic character. The authors should mention in the text that an SDW alternating $S = 0$, $S = 1$ is not realistic but rather an approximation. All that is necessary is to add, even in parentheses, a note that this is the electronic configuration in the extreme case.

2. Finally, with regards to the surface quality. I appreciate that the authors are reluctant to show such a high level of roughness. However, it is important for other thin film growers to see this quantification of roughness if they were to attempt to produce a similar study so it should really be added to the supporting information as in-situ RHEED patterns alone are not enough.

Response to Referees (3rd Submission)

We thank the Referees for their reviews, and all of their thoughtful questions and comments, which we have addressed. All changes made over the last submitted version are colored in red in the manuscript.

Summary of changes made to the main text:

1. The word “high-mobility” is changed to “high-purity” in the title.
2. Use “almost a decade in temperature” to describe the linear-T resistivity.
3. Added one sentence describing other mechanisms that may lead to linear-T resistivity.
4. Added text about the residual resistivity as a measure of sample purity, given that Matthiessen’s rule does not apply due to interplay of disorder scattering and scattering from AF quantum fluctuations.
5. Added two sentences mentioning possible connection to the recently observed superconductivity in nickelates.
6. Revised the sentence describing the $S = 1$ magnetic moment on Ni impurity sites.

Changes made to Supplementary Information:

1. Included Figs. R1 and R2 from the last response letter.
2. Added characterization of surface roughness and description of its relation to conductivity.

Reviewer #1 (Remarks to the Author):

The authors have provided a comprehensive response to my comments as well as those of Reviewers #2 and 3. I am impressed by the detail of the answers and the changes made to the manuscript/supplement. I still have some slight misgivings regarding some aspects of the interpretation/presentation, which really ought to be addressed before proceeding. But, I think the manuscript as a whole is significantly improved and I would recommend its publication. The experimental results and the timeliness of the paper will certainly interest the community.

We thank the Reviewer for the recommendation.

One thing I would insist before proceeding is that the authors include some form of the Figs. R1 and R2 shared with the referees in the supplement and/or main text. I think it is quite necessary for a reader to examine in detail the resistivity exponents for different samples, as this is the major important claim of the paper. I encourage the authors to include it in the main text, but if not there, then it should at the very least be in the supplement.

We have taken the Reviewer’s suggestions and have included Figs. R1 and R2 in the Supplementary Information and added corresponding captions.

In addition, I encourage the authors to consider my comments below, which I think would improve the manuscript.

- To address a point of Reviewer #2, maybe the authors should change the title to say “high-purity LaNiO₃ films” which is probably more precise in the context of this paper. Alternatively, where the authors state, “This sample has a ...mobility of about 160 cm²/Vs at T = 2 K. The

corresponding RRR is about 24. This is so far the highest RRR reported for LNO” if they could compare their mobility to the literature and say this is the highest mobility reported (at the same temperature), then the title of the paper would make sense.

We have revised the title of our manuscript to: “Observation of an antiferromagnetic quantum critical point in high-purity LaNiO_3 ”.

- Based on the data shown in Figs. R1 and R2 and the main text, I am still not convinced of the statement that “the resistivity $\rho(T)$ shows a linear temperature dependence over a decade of T below ~ 1.1 K in our cleanest samples.” The authors should temper their claim a bit and write “almost a decade in T” which would be a more accurate representation of the data.

We have revised the text as suggested by the Reviewer. The sentence now reads “the resistivity $\rho(T)$ shows a linear temperature dependence over almost a decade in T below ~ 1.1 K in our cleanest samples.”

- It would benefit readers if the authors would comment on/rule out other mechanisms for linear T resistivity besides the quantum critical AFM scenario. They briefly mention “Phonons are nominally not relevant in this low temperature regime given that the Debye temperature of LNO is above 400 K.” But, a sentence or two about other potential origins such as charge fluctuations, Umklapp scattering or diffusive transport would be useful.

We have added “Ni-O bond length fluctuations have also been proposed as a source of linear-T resistivity at low temperatures [Rivadulla2003], and there may be other mechanisms as well (charge fluctuations, Umklapp scatterings). However, these mechanisms are less natural for explaining our data as they would have a weak dependence on magnetic field.” into the main text.

- The authors have now written in the manuscript on page 2: “The disorder level in the sample may be characterized by the residual resistivity or RRR of the sample, assuming that the Matthiessen rule holds.” However, in Rosch’s theory, upon which much of the experimental conclusions are based, Matthiessen’s rule is expressly broken in the crossover regime where the interesting temperature dependences are found. While this may not invalidate the claim that the RRR is representative of the level of disorder, the authors should be careful in their arguments and at least revise the aforementioned statement.

We have revised the text to “The disorder level in the sample may be characterized approximately by the ‘residual’ resistivity or RRR of the sample (note that Matthiessen’s rule might not apply at $T = 2$ K due to an interplay between disorder and other scattering mechanisms).”

- Given the importance of the connection, some mentions might be made about superconductivity – both the relation of the quantum critical point to superconductivity (as pointed out by Referee #3) and with reference to the observed superconductivity infinite layer nickelates.

We have added two sentences into the main text concerning possible relation between the AFM QCP and superconductivity: “Recently, superconductivity has been observed in an infinite-layer nickelate [Li2019], which might have a connection to the QCP that we observe here in LNO. This possibility warrants further exploration.”

Reviewer #2 (Remarks to the Author):

It was very much my impression (and the impression of at least one other reviewer judging by their comments) in the initially-submitted manuscript that the authors were claiming a systematic control of disorder by cation ratio. As this has now been clarified and the possibility of oxygen vacancies has been mentioned I feel that this issue, which was my main one, has been resolved.

We thank the Reviewer for the recommendation.

However, two of my minor concerns have not been addressed satisfactorily. If these modifications are made then I believe the paper will be sufficiently complete for publication.

1. Perhaps my point concerning the electronic configuration was not clear enough. The hybridization will never be complete and the Ni-O bond will always have a mixed covalent/ionic character. The authors should mention in the text that an SDW alternating $S = 0$, $S = 1$ is not realistic but rather an approximation. All that is necessary is to add, even in parentheses, a note that this is the electronic configuration in the extreme case.

We have taken the Reviewer's suggestion, and revised the main text as: " This may be seen (approximately, due to Ni-O covalence) as a localized $3d^8$ magnetic moment ($S = 1$) at a Ni impurity site [Park2012] that is partially screened by the ligand hole which takes on the e_g symmetry of the $3d^8$ orbitals."

2. Finally, with regards to the surface quality. I appreciate that the authors are reluctant to show such a high level of roughness. However, it is important for other thin film growers to see this quantification of roughness if they were to attempt to produce a similar study so it should really be added to the supporting information as in-situ RHEED patterns alone are not enough.

We have now included the characterization of surface roughness into the Supplementary Fig 2. Correspondingly, a description regarding the surface roughness and conductivity is also added:

"We observed that although these samples have good surface crystallinity, the surface roughness is higher than those low-RRR samples grown at lower temperatures. Therefore, the sample's conductivity does not positively correlate with the sample's roughness. Figure S2d shows the characterization of the surface roughness on sample LNO_18 using x-ray reflectivity measurement. The roughness of this high-RRR sample is about 0.86 nm, which is higher than that of LNO_3 of about 0.38 nm determined from the same measurement."

We thank all the Reviewers again for their careful reading of our paper, and very constructive and insightful comments.